# Development of the First Microsatellite Multiplex PCR Panel for Meagre (*Argyrosomus regius*), a Commercial Aquaculture Species

**Antonio Vallecillos** [1] , **Emilio María-Dolores** [1], **Javier Villa** [2], **Francisco Miguel Rueda** [2], **José Carrillo** [2], **Guillermo Ramis** [3] , **Mohamed Soula** [4], **Juan Manuel Afonso** [4] **and Eva Armero** [1,*]

[1] Department of Agronomic Engineering, Technical University of Cartagena, Paseo Alfonso XIII 48, 30202 Cartagena, Spain; antonio.vallecillos@edu.upct.es (A.V.); emilio.mdolores@upct.es (E.M.-D.)

[2] Alevines del Sureste S.L., calle Cabo Cope s/n, 30880 Águilas, Spain; j.villa@avramar.eu (J.V.); f.rueda@avramar.eu (F.M.R.); j.carrillo@avramar.eu (J.C.)

[3] Department of Animal Production, University of Murcia, Avenida Teniente Flomesta 5, 30100 Murcia, Spain; guiramis@um.es

[4] Institute of Sustainable Aquaculture and Marine Ecosystems (GIA-ECOAQUA), Carretera de Taliarte s/n, 35214 Telde, Spain; mohamed@anfaco.es (M.S.); juanmanuel.afonso@ulpgc.es (J.M.A.)

* Correspondence: eva.armero@upct.es; Tel.: +34-968-325-538; Fax: +34-968-325-433

**Abstract:** In this study, a microsatellite-based multiplex PCR panel for meagre (*Argyrosomus regius*) was developed as a useful and single tool in parental assignment and population studies. Twenty-one specific and interspecific microsatellites from different aquaculture species of meagre (*Argyrosomus regius*), Japanese meagre (*A. japonicus*), red drum (*Sciaenops ocellatus*) and yellow meagre (*Acoupa weakfish*) were assessed for genetic variability, allelic range and genotype reliability. Finally, a SuperMultiplex for *Argyrosomus regius* (SMAr) was designed with only the best eight microsatellite markers. The panel assessment was performed using a batch of brood stock from one company and a sample of 616 offspring. It was possible to assign 95% of the offspring to a single pair of parents using the exclusion method. It is therefore considered an easy procedure, and a powerful and low-cost tool for parental assignment to support companies' breeding programs and to exchange information between research groups.

**Keywords:** meagre (*Argyrosomus regius*); microsatellites; parental assignment; PCR; population

## 1. Introduction

Meagre (*Argyrosomus regius*) has been one of the most important emerging fish in the Mediterranean aquaculture diversification, with a production of 12,094 tons in 2019. The main meagre producing countries were Spain (4505 tons), Turkey (3375 tons) and Greece (2415 tons) [1].

As a consequence of the growth and consolidation of the production of meagre, the industries and production companies seek the implementation of improvements in the production cycles, in order to reduce production costs and obtain a better quality final product. To achieve this, the implementation of a selective breeding program is essential. As in most fish, reproduction in meagre takes place through mass spawning, so the parents of the new individuals are not known, and this data is needed when estimating genetic parameters [2] (p. 32). Fish breeding programs are based on the identification and tracking of the individuals, both through physical tagging as well as the genetics of parent–offspring relationships by analyzing molecular markers to avoid expensive genetic analysis at all sampling points during the on-growing process [3]. Different genetic markers are used to investigate the relationship between relatives and to assist selection programs. Genome-wide distributed molecular markers, such as Single Nucleotide Polymorphisms (SNPs), are useful in both parental assignment with relatively few markers (150–200) and in genomic

selection with a large number of markers (>30 K), where they are found to improve the accuracy of estimating breeding values [4,5], and microsatellites [3], which have been widely used in parental assignment, gene mapping and population genetics studies [3,6]. A selection scheme requires designing a standardized system with a good allocation of family relationships. Different variables, such as population size and allelic diversity [7], the number of breeders to be used in spawning [8], the level of homozygosity [9] and the existence of null alleles [3], must be studied when selecting the necessary number of loci to obtain a successful assignment.

A panel of genetic markers is a set of highly polymorphic, specific and reproducible markers, assessed according to their polymorphism and their genotyping errors. Microsatellites are used as molecular markers in a panel. When combining microsatellites in a multiplex PCR, it reduces the cost, the execution times, the PCR errors and produces good efficiency in the parental assignment [3,7]. In the development of a standardized panel, it is intended to facilitate the exchange of information, the combination of data sets and the management of populations between research groups. An international panel is that which is approved and recommended by the International Society for Animal Genetics (ISAG).

In the family of the Scianids, many microsatellites have been described [10–16]. To date, in the case of meagre (*Argyrosomus regius*), no microsatellite marker panels have been developed, although several specific and interspecific microsatellites have been used in several works [10–13,17], and only a few studies [18–20] showed the multiplex they used for the parental assignment. Vallecillos et al. [21] developed a panel of markers similar to this study but with some minor modifications. Several multiplex PCRs for parental assignment have been described for gilthead sea bream [22–24], for long-snouted seahorse [25], and recently for white-leg shrimp [26] and for *Crassostrea hongkongensis* [6].

The main objective of the present study was to develop a microsatellite-based multiplex PCR panel to homogenize procedures, to enable results to be compared, to facilitate the exchange of information between laboratories and to provide a useful genetic tool to facilitate future selective breeding in the aquaculture industry.

## 2. Materials and Methods

### 2.1. Sample

Six hundred and sixteen offspring from a brood stock of nine breeders (4 males and 5 females) from Alevines del Sureste SL company were used to assess the quality of a microsatellite-based multiplex PCR panel in the parental assignment.

DNA was extracted from the caudal fin, which was preserved in ethanol, using the DNeasy kit (QIAGEN®, Hilden, Germany), and stored in a freezer (−20 °C) until the next process. Next, DNA quantity and quality were determined with a NanoDrop™ 2000 spectrophotometer v.3.7 (Thermo Fisher Scientific, Wilmington, NC, USA). The integrity of the DNA was verified by 1% agarose gel electrophoresis staining with ethidium bromide and analyzed by the photo documentation kit (AlphaDigiDoc RT2), using DNA-MARKER Beethoven (Danagen-Bioted, Badalona, Spain) as molecular weight marker.

### 2.2. Primer Desing

From the microsatellites published in the genetic map of different species of Sciaenidae, 11 microsatellites were markers of meagre, *Argyrosomus regius* [13], six of Japanese meagre, *A. japonicus* [10], three of red drum, *Sciaenops ocellatus* [14,17] and one of yellow meagre, *Cynoscion acoupa* [11]; therefore, we had 21 pairs of primers which were used to amplify microsatellites using the same PCR conditions (Table 1). The length of the amplicons ranged between 80–266 base pairs (bp), thereby ensuring a wide amplification spectrum, and optimizing the efficiency of the multiplex reaction [27]. To avoid complexity problems, a theoretical annealing temperature of 60 °C ± 2 °C was sought for all primers.

**Table 1.** Microsatellites tested for panel development in meagre (*Argyrosomus regius*).

| Authors | Species | Loci |
|---|---|---|
| Porta et al. [13] | *Argyrosomus regius* | gCT15, gA2B, CA3, CA4, CA6, CA14, CA13, CA10, gA16, gA17 and gA6 |
| Archangi et al. [10] | *Argyrosomus japonicus* | UBA853, UBA54, UBA53, UBA50, UBA6 and UBA5 |
| Turner et al. [14] | *Sciaenops ocellatus* | Soc 011 |
| Kathleen et al. [17] | *Sciaenops ocellatus* | Soc 431 and Soc 405 |
| Farias et al. [11] | *Cynoscion acoupa* | CacMic 14 |

*2.3. PCR Conditions*

Each microsatellite was individually tested using a PCR with part of the samples (12% of the sample), in order to test and confirm its correct amplification, allele size range, genetic variability and genotyping. The PCR was performed in a 12.5 µL reaction mix with the following component concentrations: 2× Type-it Microsatellites PCR kit (QIAGEN®, Hilden, Germany); 10 µmol/L for each primer; and 10 ng/µL template DNA. The thermal profile included a pre-denaturation step of 95 °C for 15 min followed by 32 cycles of denaturation-annealing-extension at 94 °C for 30 s, 60 °C for 90 s and 72 °C for 1 min, and one final elongation step at 60 °C for 30 min. Later, 1 µL of each reaction product was mixed with 6.32 µL of Hi-Di formamide and 0.36 µL of GeneScan LIZ 500 (Applied Biosystems) and amplicons were resolved by capillary electrophoresis on an ABI 3730 sequencer (Applied Biosystem, Foster City, CA, USA). The fragment size analysis software ThermoFisher MSA was used for genotyping.

*2.4. Genotyping Reliability of Microsatellite Markers*

Each microsatellite marker was assessed individually, with four quality controls being considered [3]: (A) inadequate amplification: peak height <300 RFU (relative fluorescent units); (B) null allele: preferential amplification of the shortest allele; (C) unclear band pattern: bands that make it difficult to identify between homozygous and heterozygous for adjacent alleles; (D) intermediate alleles: loci of di-nucleotides, differing from each other by 1 bp (Figure 1). The frequency of null alleles was estimated using MicroChecker software (v.2.2.3) [28], which estimated allele frequency using four different algorithms.

The concentration of the primers was adjusted until peaks between 600 and 3000 RFU were obtained, as described by Navarro et al. [23]. The PCR product was again checked on a 2% agarose gel for 45 min to test the amplification of the multiplex amplicons. As before, the amplification reading of each locus was performed under the same conditions used in the individual PCR. The electrophoregrams were analyzed by ThermoFisher MSA.

*2.5. Genetic Diversity and Parental Assignment*

Using the Cervus package (3.0.7) [29], the Hardy–Weinberg (HW) equilibrium test, the observed heterozygosity (Ho), expected heterozygosity (He) and the polymorphic information content (PIC) were studied. The markers were classified according to their PIC value as being highly informative (PIC ≥ 0.5), reasonably informative (0.5 > PIC > 0.25), or slightly informative (PIC ≤ 0.25) [3]. Wright's Fixation index ($F_{IS}$) was calculated as Mean He-Mean Ho / Mean He with the GenAlex software (v.6.0.) [30], indicating this coefficient departure from HW expectation. For the parental assignment, the exclusion method as implemented in VITASSING (v.8.2.1) software [31] was used. Breeders' effective size (Ne) was firstly estimated as $N\hat{e}_F = (4N_s N_d)/(N_s + N_d)$ [32], where $N_s$ is the number of sire breeders and $N_d$ the number of dam breeders. We also used an approach that accounts for variance in family size—$N\hat{e}_C = 4(n-2)/((K_s + V_s/K_s) + (K_d + (V_d/K_d) - 2))$ [33]—with N being the offspring sample size, $K_s$ and $K_d$ the mean numbers of offspring per sire and per dam, respectively, and $V_s$ and $V_d$ the variances of sire and dam family sizes, respectively. It is a useful approximation to the problem of estimating Ne when there are unequal contributions of breeders to offspring.

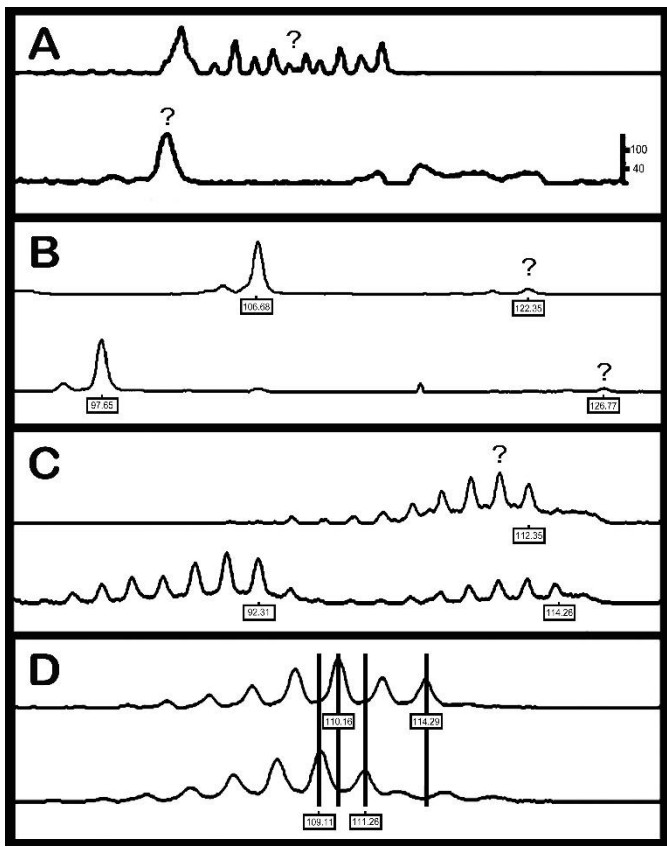

**Figure 1.** Potential errors during genotyping reliability of microsatellite markers. (**A**) Inadequate amplification, (**B**) null allele, (**C**) unclear band pattern and (**D**) intermediate alleles (source: [3]).

### 3. Results and Discussion

#### 3.1. Microsatellite Selection, Multiplex PCR Design and Parental Assignment

It is very important to perform the assessment of microsatellite markers one by one, so we did so according to the four potential errors [3]. Miquel et al. [34] reported that the reliability of genotyping is as important as the degree of polymorphism. In our study, the main problem was with low or non-existent amplification, with eight (CA6, UBA5, CA4, UBA853, CA14, UBA6, UBA54 and CA10) of the 21 microsatellite markers (38%) not amplifying—four were from *A. japonicus* and four from *A. regius*. This problem with the amplification was likely because some microsatellite markers were interspecific and/or primer design problems. Two markers had inadequate amplification (CA13 and GA16) and they could not be well observed. In addition, the production of null alleles may reduce the success of the assignment, as seen by other authors [9]; for us, two microsatellite markers showed a high rate of null alleles (21.1 and 24.3% null allele frequency for gCT15 and CA3, respectively, according to Oosterhout's estimation). Finally, a less frequent error was the appearance of intermediate alleles. Thus, only one microsatellite showed intermediate alleles (GA6); however, in Lee-Montero et al. [3], this error was the most frequent (16%).

The remaining eight microsatellites were selected as suitable (Table 2). A multiplex PCR of only these eight microsatellite markers was developed to amplify all of them in the same reaction. For this purpose, the amplicons size, different fluorochrome and the observed allele size range were considered, and the concentration for each microsatellite marker optimized after several genotyping runs (Table 2). In addition, the allele frequency for the microsatellite marker was studied and is shown in Supplementary Table S1. In this study, the amplification size was similar in all markers (from 80 to 155 bp), except in SOC011, which has a larger size than the rest, although its amplification was appropriate. The minimum distance between markers with the same fluorochrome after genotyping was 12 bp, avoiding overlaps between them. In the process of unifying different microsatellite

markers in the same PCR, it is recommended, in addition to starting from good quality DNA, to select those that have a similar size so as not to reduce the amplification of some loci [27]. However, other authors opted for microsatellites with different amplification sizes [13,23,24] to avoid overlaps between microsatellite markers dyed by the same fluorochrome.

**Table 2.** Microsatellite panel of meagre: GenBank accession number, motif, tagging fluorochrome, the primers sequence and concentration and allele size range.

| STR Loci | GenBank Accession Number | M | F | Forward Sequence (5′–3′) | Reverse Sequence (5′–3′) | Concentration (µM) | Allele Size Range |
|---|---|---|---|---|---|---|---|
| gA2B | GU724794 | $(CA)^{26}$ | 5*NED | AAGTGTGGCG TCATTTCCTCT | GTATTGATGGATAGCA AGTGTCAGA | 0.06 | 86–110 |
| UBA50 | EF462924 | $(GT)^{26}$ | 5*NED | GCACAACTGC ATCCCTTAGAT | GTTTAGAAGTGAAGA CTGCGGACTG | 0.15 | 128–152 |
| gA17 | GU724798 | $(GT)^{12}$ | 5*6-FAM | CTAGAGAAATTCATC CAGGGAAGTG | GTTTAGAGCAGAGAGTTAGCGGTTGTT | 0.06 | 80–92 |
| SOC 405 | AY161014 | $(CA)^{12}$ | 5*6-FAM | AGCCTTTTGTTTA GTTTCCCTCAT | GGGGTGTAGCAGAACCACAC | 0.06 | 112–124 |
| Cacmic 14 | DQ285034 | $(CT)_{12}$ | 5*PET | ATCTTCTCC CCTCCGTCACT | CTGTGTTGTTAAGGCGCATC | 0.06 | 132–148 |
| SOC 431 | AY161032 | $(GT)^{26}$ | 5*PET | GTGGTAGATGAAAACG TATAAAAGGAG | GTTTCATATATATAGTG TACAGCTCCAGCTTC | 0.08 | 124–148 |
| UBA53 | EF462925 | $(CA)^{14}$ | 5*VIC | TACTTCCTTCTACCCCT AAGTCTGG | GACTTTCCAGTGTAGCTGTCGTTT | 0.08 | 86–114 |
| Soc 011 | AF073258 | $(GA)^{11}$ | 5*VIC | GCCGAGTCACGAAGG AACAGAGAA | TGTCGTCTCATCTATCTCCATCTC | 0.08 | 250–266 |

M: motif of repetition; F: fluorescent tag.

In the present study, only eight microsatellite markers formed our standardized panel and were combined in an abridged SuperMultiplex for *Argyrosomus regius*, which was named "SMAr". Parental assignment was carried out with SMAr allowing for four mismatches at the first reading of the genotypes, with no prior correction for mismatches; nevertheless, most of the offspring (64.7%) were assigned with one or two mismatches to a single pair of parents (Table 3). The power of SMAr is that it allows us successfully assign 95.1% of offspring to a single pair of parents at the first genotyping reading. All the parents contributed to the spawning, although unequal contribution was observed: two out of five females produced 61.8% of the offspring, although all the females contributed to the offspring; and one out of four males contributed with 49.7% of the offspring with all of the males contributing. The effective breeding number estimation, considering only contributing males and females (Nê$_C$ = 6.9), was lower than the former (Nê$_F$ = 8.9) assumption of breeders spawned. Therefore, an initial reduction in the effective breeding number was observed. There are many authors that have developed a panel of microsatellite markers for the management of the pedigree of the study populations and to support breeding programs [3,25,26]. Nousias et al. [18], for meagre with a 10-microsatellites panel, reached an overall 91.1% parental assignment; Lee-Montero et al. [3], for gilthead seabream, used two panels with 11 markers in each (SMsa1 and SMsa2) with a 100% assignment; and Vandeputte et al. [31], for rainbow trout with a 5-microsatellites panel, showed 57.1% assignment to a single correct and 40.3% to multiple pairs of parents at the first reading of genotypes, using the same VITASSING software. Ren et al. [26], for white-leg shrimp, used two panels with 6 markers in each (SMpv01 and SMpv02) that were tested using CERVUS and COLONY with a 100 and 94.3% assignment, respectively; and López et al. [25], for long-snouted seahorse, used a panel with 6 markers, that were tested using CERVUS and FAP with 100 and 95.8% assignment, respectively. Previous studies did not point out the number of readings of genotypes; therefore, we suppose that the shown parental assignment percentage is after genotype corrections. Only Vandeputte et al. [31] highlighted that the assignment was at the first reading of the genotype.

**Table 3.** Number of offspring assignment from parental pairs identified with eight microsatellites when four mismatches are allowed, in 616 fish.

| | Maximum Number of Mismatches Allowed | | | | |
|---|---|---|---|---|---|
| Kind of assignment observed | 0 | 1 | 2 | 3 | 4 |
| Single pair of parents | 83 | 240 | 399 | 545 | 586 |
| Multiple pairs of parents | 0 | 5 | 7 | 1 | 2 |
| Unassigned | 533 | 371 | 210 | 70 | 28 |

Single incorrect assignment was zero regardless of the number of mismatches allowed.

### 3.2. Genetic Diversity

Genetic diversity, analyzed by the heterozygosity and polymorphism degree of each microsatellite, FIS and HW equilibrium is shown in Table 4. In our population of offspring, for the selected microsatellite markers, the mean values and the range were: for the number of alleles, 7.9 (from 5 to 13); for observed heterozygosity (Ho), 0.69 (from 0.35 to 0.97); for expected heterozygosity (He), 0.61 (from 0.32 to 0.81); for polymorphism information content (PIC), 0.57 (from 0.35 to 0.80); and for $F_{IS}$, −0.12 (from 0.03 to −0.22).

**Table 4.** Genetic diversity of the offspring population using the optimized SuperMultiplex *Argyrosomus regius* (SMAr).

| Meagre STR Loci | Nº Alleles | Ho | He | PIC | $F_{IS}$ | *p*-Value HW |
|---|---|---|---|---|---|---|
| gA2B | 8 | 0.62 | 0.52 | 0.46 | −0.19 | *** |
| UBA50 | 13 | 0.97 | 0.81 | 0.80 | −0.21 | *** |
| gA17 | 7 | 0.73 | 0.69 | 0.68 | −0.07 | *** |
| SOC 405 | 5 | 0.35 | 0.32 | 0.35 | −0.10 | *** |
| Cac mic 14 | 8 | 0.79 | 0.72 | 0.70 | −0.09 | *** |
| SOC 431 | 8 | 0.61 | 0.50 | 0.57 | −0.22 | *** |
| UBA53 | 9 | 0.58 | 0.60 | 0.47 | 0.03 | NS |
| Soc 011 | 9 | 0.88 | 0.80 | 0.77 | −0.10 | *** |

Ho: observed heterozygosity; He: expected heterozygosity; PIC: polymorphism information content; $F_{IS}$: fixation index; HW: Hardy-Weinberg equilibrium; *** *p*-value $\leq$ 0.001 that implies disequilibrium of HW. NS: not significant.

In our population, the mean Ho was high, and the mean He was slightly lower for most microsatellites, except SOC405 and UBA53, which showed a lower Ho because one or two alleles showed a much higher frequency than the others (Table S1). Other studies have observed slightly less heterozygosity [35] (pp. 21–24) than in our study, which is probably due to the fact that we have chosen highly polymorphic microsatellites and, although the number of breeders was very small in our work, all of them contributed and have never been subjected to a selection process. In other aquaculture species such as gilthead sea bream, something similar occurs, since they are stocks that have not undergone any breeding program [3,23], with a high population heterozygosity. Most of the microsatellites showed a significant HW disequilibrium, which revealed an excess of heterozygotes, perhaps due to the small effective number of breeders and their unequal contribution [36]; this is reinforced by the lower Nê$_C$ estimated when we applied the Crow and Denniston [33] approach considering the true variance in family size.

A microsatellite-based multiplex PCR panel has to be sufficiently informative to ensure high quality results [37]. In our work, five out of eight microsatellites were highly informative and three were reasonably informative (gA2B, SOC405 and UBA53), a similar result to that of the multiplex proposed by Lee-Montero et al. [3], although the level of polymorphism of the microsatellite markers also depends on the population [38]. In addition, the use of non-coding regions in markers favors their genetic variability not being lost over time, especially when they are used in breeding programs and some selection pressure is exerted on them [3] so that mutation and drift are the main factors driving allele frequency changes within and among populations (although draft due to close linkage

to coding regions has been suggested [39]). For coding markers, Chistiakov et al. [40] observed how their variability was reduced in successive selected generations.

Future selective breeding for traits associated with aquaculture production efficiency is greatly desired in the aquaculture industry. However, one of the main concerns in a selection process is the remarkably reduced number of the effective population size that can lead to an increase in inbreeding. Indeed, we observed significantly unequal breeder contributions to the offspring. This outcome seriously affected the $N\hat{e}_C$ estimates, which decreased compared with $N\hat{e}_F$; therefore, inbreeding is likely to increase in subsequent generations. Microsatellite markers can help us lead mating to prevent further inbreeding.

A microsatellite-based multiplex PCR panel is a useful and low-cost tool to enhance genetic breeding programs through stock management, individual genetic identification and pedigree reconstruction, and in genetics studies of populations. In this case, the panel has been designed to support and carry out the first genetic improvement program in meagre (GENECOR 2020) in Spain. The application of SNP genetic markers to breeding is particularly valuable for difficult traits or those that cannot be measured in the selection candidates (e.g., disease resistance and fillet quality), in which the accuracy of the estimated breeding values can improve [4,5]. However, for other traits associated with aquaculture production with medium-high heritability, such as growth [21], the accuracy for estimated breeding values using a pedigree-based approach or genomic prediction is expected to be similar [41]. Furthermore, despite the fact that the cost of SNP genotyping has been dramatically reduced in the last few years, the cost of genotyping with microsatellites (8–10 markers) has been estimated to be two to three times cheaper compared to SNPs (150–200 markers) when they are used for parental assignments: EUR 3 for microsatellites versus EUR 7–11 for SNPs, not including DNA extraction.

## 4. Conclusions

The design of microsatellite markers SuperMultiplex for *Argyrosomus regius* (SMAr) that amplifies all loci in one reaction is a way to standardize procedures and to share results among researchers and companies. The use of this panel is a powerful and economic tool in parental assignment and to support a breeding program of great interest to the aquaculture industry.

**Supplementary Materials:** The following supporting information can be downloaded at: https://www.mdpi.com/article/10.3390/fishes7030117/s1, Table S1: Allele frequency of the offspring population, in 616 fish.

**Author Contributions:** Conceptualization, A.V. and E.A.; methodology, A.V.; software, A.V.; validation, G.R. and E.A.; formal analysis, A.V.; investigation, A.V.; resources, E.A.; data curation, A.V.; writing—original draft preparation, A.V.; writing—review and editing, E.A.; visualization, E.M.-D., J.V., F.M.R., J.C., G.R., M.S. and J.M.A.; supervision, E.A.; project administration, E.A.; funding acquisition, E.A. and J.V. All authors have read and agreed to the published version of the manuscript.

**Funding:** This research was funded by Fundación Séneca-Agencia de Ciencia y Tecnología de la Región de Murcia through the project "Mejora de la competitividad del sector de la corvina a través de la selección genética (GENECOR, 21002/PI/18)" in the call for grants for projects for the development of scientific and technical research by competitive groups, included in the Regional Program for the Promotion of Scientific and Technical Research (Action Plan 2018). A.V. was funded by a pre-doctoral research fellowship (20716/FPI/18. Fundación Séneca. Cofinanciado por grupo Avramar S.L. Región de Murcia (Spain)). The Avramar S.L. group provided financial support and the use of its facilities for this study.

**Institutional Review Board Statement:** The animal study protocol was approved by the Ethics Committee Polytechnic University of Cartagena of the Region of Murcia (protocol code CEI21_006 and 27 May 2021).

**Informed Consent Statement:** Not applicable.

**Data Availability Statement:** Data sharing is absent, and thus, data are not needed.

**Conflicts of Interest:** The authors declare no conflict of interest.

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
