# Peer review of "Development of the First Microsatellite Multiplex PCR Panel for Meagre (Argyrosomus regius), a Commercial Aquaculture Species"

_fishes, doi:10.3390/fishes7030117_

Round 1

Reviewer 1 Report

Journal: Fishes (ISSN 2410-3888)

Manuscript ID: fishes-1720163

Type: Communication

Title: Development of a microsatellites based multiplex PCR standardized panel for Meagre (Argyrosomus regius)

Revision fishes-1720163

In this manuscript, the authors implemented the first multiplex PCR panel with microsatellites for meagre (A. regius), a commercial aquaculture species. It is an interesting communication, and no major flaws was detected in Materials and Methods. Nevertheless, there are some important pending issues that need to be addressed before the manuscript can be consider for publication, which are listed below.

Title

  • Lines 2-3: I would recommend emphasizing the goal achieved and the research context in the title. Maybe, something like this:

“Development of the first microsatellite multiplex PCR panel for meagre (Argyrosomus regius), a commercial aquaculture species”.

Abstract

  • Lines 17-18: The common name “meagre” is used for two different species: regius and A. japonicus. This happens throughout the manuscript, being confusing. According to FAO, A. japonicus would correspond to the common name of “Japanese meagre”. You can check this at FAO ASFIS List of Species for Fishery Statistics Purposes: https://www.fao.org/fishery/en/collection/asfis/en

Introduction

—Lines 29-32: I agree with the use of 2019 data, the year before COVID-19 pandemic. Nevertheless, I would recommend the Global aquaculture production database (FAO), a more traceable data source:  https://www.fao.org/fishery/statistics-query/en/aquaculture. There are some differences in production (tons). Furthermore, I consider necessary to include the economic value of this aquaculture production, see: https://www.fao.org/fishery/statistics-query/en/aquaculture/aquaculture_value.

—Lines 48-50: the clarity of these two sentences could be improved. Consider changing “A panel is” by “A panel of genetic markers is”.  In the second sentence, maybe something like this: “Different genetic markers are used in these genetic tools, such as: Single Nucleotide Polymorphism (SNPs; [REF], Restriction Fragment Length Polymorphisms (RFLPs; [REF], microsatellites [REF]…”. 

—Line 56: Other genetic markers were described in Scianid family for commercial purposes?

—Line 66: Consider changing “useful tool” by “useful genetic tool”.

Material and methods

This section presents a clear and detailed reading. There are a few aspects that need further explanations. The main drawback would be the low resolution of Figure 1. It is very difficult to read the numbers inside the black squares with the printed manuscript.

Below is a summary of specific comments:

—Line 74: The degree Celsius symbol is ºC (https://doi.org/10.6028/NIST.SP.330-2019). Use the correct symbol throughout the manuscript.

—Lines 87-89: Please clarify this sentence, I do not understand well the meaning of “theoretical annealing temperature of 60 ºC ± 2 ºC”.  In line 100 is specified that annealing temperature in PCR was 60 ºC (a “consensus” annealing temperature for the set of primers, I understood). Then, optimum temperatures for the 21 pairs of primers would be between 58-62 ºC?

—Line 94: What approximate percentage of the sample size would represent "part of the samples"? Please, consider including some number range or percentage.

—Line 105: I can not see anything with this link, login user is required. The same inconvenience at line 122. If this inconvenience was not arranged, I recommend removing both links, since they do not provide additional information for the reader.

—Line 110: “amplification of the short allele” is correct? Maybe the “shortest allele”?

—Line 130: change “VITASSING (v.8_2.1)” by “VITASSING (8.2.1)”.

Results and discussion

The Results and discussion section is generally well written.

—Line 138: Please clarify/expand the explanation for the cause of microsatellites with low or non-existent amplification (8). It is noteworthy that all of them belong to A. regius (the species sampled) and its congeneric species (A.japonicus).

—Line 142: How high was the rate of null alleles?

—Lines 147-149: This sentence should be divided into two parts: (i) “The remaining eight microsatellites were selected as suitable (Table 2)”. The second part, “shows the name and GenBank accession number for each locus, the motif for each microsatellite […]; should be included in some way in the header or in the Table 2 itself (e.g., change “Accession” by “GenBank accession number” in Table 2).

—Line 154: “similar” instead “homogeneous”.

—Line 174: “most of the offspring were assigned with one or two mismatches”. Please, include the percentage.

—Lines 175-176: “SMAr allowed us to successfully assign 95% of offspring to a single pair of parents, where all of the parents contributed to the spawning.” This result has caught my attention, due to the offspring sample size (N = 616) were (theoretically) from nine breeders (4 males and 5 females). What assignment had the remaining 5%? Was an expected result, why? If authors considered, it would be interesting to add some additional information about this issue.

—Line 177: I would recommended change the subsection name by “Genetic information”. “Genetic structure” is not very related to the content.

—Lines 178-180: This paragraph seems disconnected from the rest of this subsection. I would suggest removing it.

—Lines 184-185: The lower value of the observed and expected heterozygosity ranges in the main text do not seem to correspond to those in Table 3. Please review it.

—Line 195: What was the p-value threshold? Please, include it.

—Lines 224-226: It would be advisable to add citations related to this sentence.

Conclusions.

In this manuscript is mentioned the economic advantages of this genetic tool, several times (e.g., “low-cost tool”).

How much would it cost approximately a 96-well sample plate using this microsatellite panel? I consider this question very interesting for the readers.

—Line 230: “economic tool” instead of “economical tool”.

References

—Line 256: I would recommend FAO database (https://www.fao.org/fishery/statistics-query/en/aquaculture).

—Line 356: Publication year is not in bold characters.

Author Response

In this manuscript, the authors implemented the first multiplex PCR panel with microsatellites for meagre (A. regius), a commercial aquaculture species. It is an interesting communication, and no major flaws was detected in Materials and Methods. Nevertheless, there are some important pending issues that need to be addressed before the manuscript can be consider for publication, which are listed below.

Title

  • Lines 2-3: I would recommend emphasizing the goal achieved and the research context in the title. Maybe, something like this:

“Development of the first microsatellite multiplex PCR panel for meagre (Argyrosomus regius), a commercial aquaculture species”.

Thanks for your contribution

Abstract

  • Lines 17-18: The common name “meagre” is used for two different species: regius and A. japonicus. This happens throughout the manuscript, being confusing. According to FAO, A. japonicus would correspond to the common name of “Japanese meagre”. You can check this at FAO ASFIS List of Species for Fishery Statistics Purposes: https://www.fao.org/fishery/en/collection/asfis/en

Corrected, thanks for this information.

Introduction

—Lines 29-32: I agree with the use of 2019 data, the year before COVID-19 pandemic. Nevertheless, I would recommend the Global aquaculture production database (FAO), a more traceable data source:  https://www.fao.org/fishery/statistics-query/en/aquaculture. There are some differences in production (tons). Furthermore, I consider necessary to include the economic value of this aquaculture production, see: https://www.fao.org/fishery/statistics-query/en/aquaculture/aquaculture_value.

Corrected and added

—Lines 48-50: the clarity of these two sentences could be improved. Consider changing “A panel is” by “A panel of genetic markers is”.  In the second sentence, maybe something like this: “Different genetic markers are used in these genetic tools, such as: Single Nucleotide Polymorphism (SNPs; [REF], Restriction Fragment Length Polymorphisms (RFLPs; [REF], microsatellites [REF]…”. 

Corrected and added, thanks for your contribution

—Line 56: Other genetic markers were described in Scianid family for commercial purposes?

Yes, since we cited articles that have described microsatellite markers, which will have a commercial application. likewise, in the study by Lee-Montero et al. 2013 (doi:10.1111/age.12037) and in ours, they have a commercial purpose.

—Line 66: Consider changing “useful tool” by “useful genetic tool”.

Corrected, thanks for your point.

Material and methods

This section presents a clear and detailed reading. There are a few aspects that need further explanations. The main drawback would be the low resolution of Figure 1. It is very difficult to read the numbers inside the black squares with the printed manuscript.

Substitution of the original figure 1, resulting in a better resolution. 

Below is a summary of specific comments:

—Line 74: The degree Celsius symbol is ºC (https://doi.org/10.6028/NIST.SP.330-2019). Use the correct symbol throughout the manuscript.

Corrected, thanks for this information.

—Lines 87-89: Please clarify this sentence, I do not understand well the meaning of “theoretical annealing temperature of 60 ºC ± 2 ºC”.  In line 100 is specified that annealing temperature in PCR was 60 ºC (a “consensus” annealing temperature for the set of primers, I understood). Then, optimum temperatures for the 21 pairs of primers would be between 58-62 ºC?

The sentence was not clear, and lead to confusion. For this reason, we have changed this sentence “To avoid complexity problems, a theoretical annealing temperature of 60 ºC ± 2 ºC was sought for all primers”

Regarding your question; yes, the temperature range of our primers ranged from 58-62 ºC.

—Line 94: What approximate percentage of the sample size would represent "part of the samples"? Please, consider including some number range or percentage.

Added

—Line 105: I can not see anything with this link, login user is required. The same inconvenience at line 122. If this inconvenience was not arranged, I recommend removing both links, since they do not provide additional information for the reader.

I have checked both links, and yes, they can be viewed correctly. To access the app, you have to create a user in the cloud and then you can log into MSA. Finally, we have decided to delate this link.

—Line 110: “amplification of the short allele” is correct? Maybe the “shortest allele”?

Corrected

—Line 130: change “VITASSING (v.8_2.1)” by “VITASSING (8.2.1)”.

Corrected

Results and discussion

The Results and discussion section is generally well written.

—Line 138: Please clarify/expand the explanation for the cause of microsatellites with low or non-existent amplification (8). It is noteworthy that all of them belong to A. regius (the species sampled) and its congeneric species (A.japonicus).

Thanks for your comment. We have expanded a little bit explanation: “In our study, the main problem was in the low or non-existent amplification, eight (CA6, UBA5, CA4, UBA853, CA14, UBA6, UBA54, and CA10) of the 21 microsatellite markers (38%) did not amplify; four were from A. japonicus and four from A. regius; this problem with the amplification was probably because four microsatellite markers were interspecific and/or primers design problems”

—Line 142: How high was the rate of null alleles?

To check for the presence of null alleles in these microsatellites (gCT15 and CA3), we used the MicroChecker software (2.2.3), in which, according to Oosterhout, the frequency of null alleles for GCT15 was 21.1% and for CA 24.3%. We have added MicroChecker software in M&M and in result this sentence line 144-145: “21.1 and 24.3% null allele frequency for gCT15 and CA3 respectively, according to Oosterhout’s estimation”

—Lines 147-149: This sentence should be divided into two parts: (i) “The remaining eight microsatellites were selected as suitable (Table 2)”. The second part, “shows the name and GenBank accession number for each locus, the motif for each microsatellite […]; should be included in some way in the header or in the Table 2 itself (e.g., change “Accession” by “GenBank accession number” in Table 2).

Corrected, thank you for your objection

—Line 154: “similar” instead “homogeneous”.

Corrected

—Line 174: “most of the offspring were assigned with one or two mismatches”. Please, include the percentage.

Added. In addition, we have included the table 3 with the number of offspring assignment from parental pairs taking into account the number of mismatches.

—Lines 175-176: “SMAr allowed us to successfully assign 95% of offspring to a single pair of parents, where all of the parents contributed to the spawning.” This result has caught my attention, due to the offspring sample size (N = 616) were (theoretically) from nine breeders (4 males and 5 females). What assignment had the remaining 5%? Was an expected result, why? If authors considered, it would be interesting to add some additional information about this issue.

The fact that this is a highly heterozygous population, which has not been subjected to a selection program, facilitates the assignment of parents, making it successful and simple.

In response to your comment, we have added a table showing the type of contribution in the parent assignment, based on the number of mismatches.

With respect to your question, the remaining 5% corresponds to individuals who were assigned to multiple parental pairs (2.6%) and those who were not assigned (2.4%).

—Line 177: I would recommended change the subsection name by “Genetic information”. “Genetic structure” is not very related to the content.

Corrected and modified, taking into account the comments of all the reviewers of this subtitle.

—Lines 178-180: This paragraph seems disconnected from the rest of this subsection. I would suggest removing it.

Corrected

—Lines 184-185: The lower value of the observed and expected heterozygosity ranges in the main text do not seem to correspond to those in Table 3. Please review it.

Corrected. there has been a typing problem, and the heterozygosity for the SOC405 marker was put wrong in the table.

—Line 195: What was the p-value threshold? Please, include it.

Corrected and added in table 4.

—Lines 224-226: It would be advisable to add citations related to this sentence.

Added

Conclusions.

In this manuscript is mentioned the economic advantages of this genetic tool, several times (e.g., “low-cost tool”).

How much would it cost approximately a 96-well sample plate using this microsatellite panel? I consider this question very interesting for the readers.

In view of your comment, we have added a sentence about the average price of genotyping with microsatellite markers, Line 246-249: “Furthermore, the cost of genotyping with microsatellites (8-10 markers) was estimated in our laboratory to be two to three times cheaper compared to SNPs (150-200 markers) when are used for parental assignments, 3 € for microsatellites versus 7-11 € for SNPs, not including DNA extraction.”

—Line 230: “economic tool” instead of “economical tool”.

Corrected

References

—Line 256: I would recommend FAO database (https://www.fao.org/fishery/statistics-query/en/aquaculture).

Corrected

—Line 356: Publication year is not in bold characters.

Corrected

Reviewer 2 Report

This manuscript deals with the optimization of a PCR multiplex to genotype microsatellite loci of the meagre (Argyrisomus regius) from available microsatellites already described in this and related Sciaenidae species. As there are not “de novo” microsatellite loci in the paper, the authors should justify why their multiplex with only 8 microsatellite loci is better than the previously available and indicated in references cited in lines from 56 to 59. For instance the same authors published a paper on this species using 10 microsatellites (the 8 of the present paper and two more) (reference #18, Vallecillos et al. 2021. Animals 11, 3285), and Nousias et al in 2021 published a multiplex with 12 loci in the Fishes -mdpi journal (reference #16, Nousia et al. 2021. Fishes 6, 78). The only significant PCR modification is that they now used 60ºC instead of 57ºC as annealing temperature, why this modification? In addition, they should better present, and discuss, the results of the assignment of the offspring to parents from 0 mismatches to only 1 o 2. The four mismatches allowed in their analyses, looks as excessive for a set of 8 loci.

Specific comments:

1-Please check the authors’ names. Valleicllos, should be Vallecillos.

2-Sentence in line 57 should be adjusted. As commented before, there are available several panels of microsatellite loci for the meagre species analysed in this manuscript

3- Line 78 DNAMARKER Beeethoven should be Beethoven.

4- Paragraph in lines 108-112 is somewhat awkward. Maybe present as quality controls instead of errors: adequate amplification (RFU>300); without null alleles; clear band patterns and pure repeats.

5- Section 2.5 is related with Genetic diversity rather than with genetic structure. I suggest modifying to: 2.5 Genetic diversity and parental assignment. Please in the results section, and Table 3, dealing with genetic diversity analysis, describe separately genetic variation for parents and offspring. Also clarify if conformance to HW equilibrium refers to the 616 meagres from the offspring or to the pool of parents and their offspring. (I guess that pooling the two groups is senseless). Add a FIS (and significance)  column(s)  in Table 3 to indicate which loci agree to HW genotype expectations. 

6-Figure 2 looks as reiterative with column “allele size range” in Table 3. It could be removed to gain space to present assignment results.

7-As commented assignment results for the 616 offspring fishes to parents (4 sires and 5 dams) should be presented in the manuscript. I suggest a table such as:

0 mismatch

1 mismatch

S1

S2

S3

S4

Not assigned

sire

S1

S2

S3

S4

Not assigned

sire

D1

D1

D2

D2

D3

D3

D4

D4

D5

D5

Not assigned Dam

Not assigned Dam

Or, alternatively, a table similar to Table 1 in ref#28 (Vandeputten et al 2006. Molecular ecology notes, 6, 265-267).

8- Sentence in lines 194 to 196, refering to HW disequilibrium for the SOC405 locus needs clarification. Why fewer alleles resulted in a significant departure of HW genotype expectations?. A positive and significant FIS value (please add FIS values in Table 3) may reflect a null allele at this locus.  

9-Sentence in line 202-204 also needs clarification. Microsatellite loci are often considered neutral markers, so mutation and drift are the main factors driving allele frequency changes within and among populations, but sometimes hitchhiking due to close linkage to coding regions has been suggested (e.g. Hansen et al. 2007. Molecular Ecology 16, 1413-1425).

Author Response

This manuscript deals with the optimization of a PCR multiplex to genotype microsatellite loci of the meagre (Argyrisomus regius) from available microsatellites already described in this and related Sciaenidae species. As there are not “de novo” microsatellite loci in the paper, the authors should justify why their multiplex with only 8 microsatellite loci is better than the previously available and indicated in references cited in lines from 56 to 59. For instance the same authors published a paper on this species using 10 microsatellites (the 8 of the present paper and two more) (reference #18, Vallecillos et al. 2021. Animals 11, 3285), and Nousias et al in 2021 published a multiplex with 12 loci in the Fishes -mdpi journal (reference #16, Nousia et al. 2021. Fishes 6, 78). The only significant PCR modification is that they now used 60ºC instead of 57ºC as annealing temperature, why this modification? In addition, they should better present, and discuss, the results of the assignment of the offspring to parents from 0 mismatches to only 1 o 2. The four mismatches allowed in their analyses, looks as excessive for a set of 8 loci.

In relation to your previous comment, I would like to clarify that this article is an optimization of this panel, which was already published in a previous article as you have commented (Vallecillos et al 2021), but in that case this panel was not optimized, the concentrations were not shown, two markers have been removed and the colors of the fluorochromes have been changed. Also, Nousias et al. 2020 do not show primers concentration.

Regarding annealing temperature, line 94-96 has been modified: "To avoid complexity problems, a theoretical annealing temperature of 60 °C ± 2 °C was sought for all primers", to avoid confusion.

Finally, we have proceeded to modify the explanation of parental assignment, as recommended by you for the four possible mismatches. To this end, we have included in the results and discussion section table 3 “Number of offspring assignment from parental pairs identified with eight microsatellites when four mismatches are allowed, in 616 fish.”

Specific comments:

1-Please check the authors’ names. Valleicllos, should be Vallecillos.

Checked thanks.

2-Sentence in line 57 should be adjusted. As commented before, there are available several panels of microsatellite loci for the meagre species analysed in this manuscript

3- Line 78 DNAMARKER Beeethoven should be Beethoven.

Checked thanks.

4- Paragraph in lines 108-112 is somewhat awkward. Maybe present as quality controls instead of errors: adequate amplification (RFU>300); without null alleles; clear band patterns and pure repeats.

Corrected

5- Section 2.5 is related with Genetic diversity rather than with genetic structure. I suggest modifying to: 2.5 Genetic diversity and parental assignment. Please in the results section, and Table 3, dealing with genetic diversity analysis, describe separately genetic variation for parents and offspring. Also clarify if conformance to HW equilibrium refers to the 616 meagres from the offspring or to the pool of parents and their offspring. (I guess that pooling the two groups is senseless). Add a FIS (and significance) column(s) in Table 3 to indicate which loci agree to HW genotype expectations. 

The title of this section, we have changed it. we have written “Genetic diversity” because parental assignment was included in the previous section.

Regarding results, is refers only to the offspring. We have considered that the information of the parents was not significant because the reduced number of parents.

Also, the main aim of the paper is the design of the microsatellite panel, not the study of the population.

Here are some of the results in case you find them interesting: “In the population of parents, for the selected microsatellite markers the mean values and the range were: for number of alleles 4.5 (from 3 to 7), for observed heterozygosity (Ho) 0.97 (from 0.87 to 1), for expected heterozygosity (He) 0.68 (from 0.51 to 0.82) and for polymorphism information content (PIC) 0.57 (from 0.35 to 0.80)”

Besides, we have included in the table 4 the value of Fis (calculated with GenAlex, added in M&M now) and significance of HW equilibrium (Calculated with Cervus).

6-Figure 2 looks as reiterative with column “allele size range” in Table 3. It could be removed to gain space to present assignment results.

Corrected and your comment has been considered 

7-As commented assignment results for the 616 offspring fishes to parents (4 sires and 5 dams) should be presented in the manuscript. I suggest a table such as:

0 mismatch

1 mismatch

S1

S2

S3

S4

Not assigned

sire

S1

S2

S3

S4

Not assigned

sire

D1

D1

D2

D2

D3

D3

D4

D4

D5

D5

Not assigned Dam

Not assigned Dam

Or, alternatively, a table similar to Table 1 in ref#28 (Vandeputten et al 2006. Molecular ecology notes, 6, 265-267).

Added. Based on your comment, we have decided to add table 3, to make it clearer what the family's contribution was according to the number of mismatches.

8- Sentence in lines 194 to 196, refering to HW disequilibrium for the SOC405 locus needs clarification. Why fewer alleles resulted in a significant departure of HW genotype expectations?. A positive and significant FIS value (please add FIS values in Table 3) may reflect a null allele at this locus.  

The study of HW equilibrium was not the aim of the article, and we have not realized about this unfortunate mistake. But, thanks to your comment, we have calculated Fis with GenAlex software and realized that most of microsatellites were in HW desequilibirum. We have changed that sentence, line 209-211: “Most of the microsatellites showed a significant HW disequilibrium, which revealed an excess of heterozygotes, maybe due to the small effective number of breeders and their unequal contributions (Garcia-Celdran et al  2016)”

Taking into account your comment, we have added the value of FIS and the P-value of HW in Table 4.

9-Sentence in line 202-204 also needs clarification. Microsatellite loci are often considered neutral markers, so mutation and drift are the main factors driving allele frequency changes within and among populations, but sometimes hitchhiking due to close linkage to coding regions has been suggested (e.g. Hansen et al. 2007. Molecular Ecology 16, 1413-1425).

We have tried to be more explicative. Thanks

Round 2

Reviewer 1 Report

Revision fishes-1720163

In this new version of the manuscript, the authors have implemented all changes recommended in the first revision, with additional information. All questions were adequately answered.

Material and Methods

Line 96: Typo with the degrees Celsius symbol.

Author Response

Revision fishes-1720163

In this new version of the manuscript, the authors have implemented all changes recommended in the first revision, with additional information. All questions were adequately answered.

Material and Methods

Line 96: Typo with the degrees Celsius symbol.

Thanks for your comment, corrected.

Reviewer 2 Report

This revised version of the manuscript has incorporated some of my comments to the previous one. However, the authors have not explained yet why the designed multiplex is better than the already available. They have reduced the number of microsatellite loci that certainly is a cost-effective improvement for rutinary studies on meagre, but apparently their parental assignments are worse than the one reported by Nousias et al. (2020) paper. These authors using a panel of 10 microsatellite, the same VITASIGN software and limiting to a single mismatch, properly identify the 87-95% of parents of two commercial batches of meagre aquaculture. In the present manuscript the rate for correct assignment of parents with just one mismatch was around 40% (Table 3). This is somewhat surprising, since the levels of diversity (Number of alleles) should produce better assignment results according to (Vandeputte et al., 2006). Altogether suggesting that some alleles are present in low frequency (He and PIC results points in this way) or there may be a high number of genotyping errors in this study (close to 10%). How many times have each fish genotyped in the present study to estimate genotype errors?

The paper may be improved  by better presenting  and discussing their assignment results in comparison with the dams and sires used to obtain the analysed offspring. 

Additional comments:

  • In paragraph from line 89 to 95 species names should be in italics and use Cynoscion acoupa as scientific name for the acoupa weakfish
  • Table 3 should be complete as in the Vandeputte et al (2006) manuscript by adding a row for single incorrect and other for unassigned.
  • Sentence in line 213-215 could be reinforced by calculating effective size of the offspring for instance from Linkage disequilibrium (NE estimator softwre) and comparing with NE= (4 + Nf * Nm)/(Nf + Nm) where Nf are the dams and Nm the sires
  • Maybe an allele frequency table should be added as supplementary material of the manuscript.
  • Discussion must be improved to compare own results with those available from literature

References

Nousias O, Tsakogiannis A, Duncan N, Villa J, Tzokas K, Estevez A, et al. Parentage assignment, estimates of heritability and genetic correlation for growth-related traits in meagre Argyrosomus regius. Aquaculture 2020; 518.

Vandeputte M, Mauger S, Dupont-Nivet M. An evaluation of allowing for mismatches as a way to manage genotyping errors in parentage assignment by exclusion. Molecular Ecology Notes 2006; 6: 265-267.

Author Response

This revised version of the manuscript has incorporated some of my comments to the previous one. However, the authors have not explained yet why the designed multiplex is better than the already available. They have reduced the number of microsatellite loci that certainly is a cost-effective improvement for rutinary studies on meagre, but apparently their parental assignments are worse than the one reported by Nousias et al. (2020) paper. These authors using a panel of 10 microsatellite, the same VITASIGN software and limiting to a single mismatch, properly identify the 87-95% of parents of two commercial batches of meagre aquaculture. In the present manuscript the rate for correct assignment of parents with just one mismatch was around 40% (Table 3). This is somewhat surprising, since the levels of diversity (Number of alleles) should produce better assignment results according to (Vandeputte et al., 2006). Altogether suggesting that some alleles are present in low frequency (He and PIC results points in this way) or there may be a high number of genotyping errors in this study (close to 10%). How many times have each fish genotyped in the present study to estimate genotype errors?

Based on your previous comment, we are not saying that our multiplex is better than those already mentioned in previous studies, but that ours is the first to publish the concentration of the loci, so that it can be a reproducible panel for other authors, and the allelic range in our population to give more information.

As for the percentage of assignments with a single mismatch, those represented in the table 3 would be for the first run of genotypes, once the appropriate modifications of the genotype are made, this percentage increased to 98.2%. Besides, we have considered “The power of SMAr is that it allows us to obtain a successfully assign 95.1% of offspring to a single pair of parents at the first genotyping run”, we have emphasized this point in the manuscript (Line 180-182). We have obtained similar results (64.7% of the offspring were assigned with one or two mismatches to a single pair of parents) to Vandeputte et al 2006 who observed “57.1% of correct unique assignments in the ‘real life’ example with rainbow trout allowing for two mismatches”.

In this study, fish were genotyped several times, depending on which micros were included or excluded from the primer mix, but without genotypes correction. If we consider genotyping errors as mismatch then it is 7.5%.

Additional comments:

  • In paragraph from line 89 to 95 species names should be in italics and use Cynoscion acoupa as scientific name for the acoupa weakfish

Thanks for your comment, corrected.

  • Table 3 should be complete as in the Vandeputte et al (2006) manuscript by adding a row for single incorrect and other for unassigned.

We included those unassigned as a footnote to the table but following the recommendations of their commentary we have decided to include unassigned in the table, and to do so as Vandeputte et al (2006). For single incorrect (SI), we did not have SI, then we have decided to include this information as a footnote.

  • Sentence in line 213-215 could be reinforced by calculating effective size of the offspring for instance from Linkage disequilibrium (NE estimator software) and comparing with NE= (4 + Nf * Nm)/(Nf + Nm) where Nf are the dams and Nm the sires

In accordance with your comment, we have reinforced the comment on lines 231-233 by calculating the effective population size using two methods, Falconer's (Nef) and Crow and Denniston's (Nec) that considers unequal contribution, which we have included in M&M.

  • Maybe an allele frequency table should be added as supplementary material of the manuscript.

Added with the attached file, thank you for your comment because we have added extra information to improve the communication.

  • Discussion must be improved to compare own results with those available from literature

We have tried to improve the discussion with both papers.

Nousias O, Tsakogiannis A, Duncan N, Villa J, Tzokas K, Estevez A, et al. Parentage assignment, estimates of heritability and genetic correlation for growth-related traits in meagre Argyrosomus regius. Aquaculture 2020; 518.

Vandeputte M, Mauger S, Dupont-Nivet M. An evaluation of allowing for mismatches as a way to manage genotyping errors in parentage assignment by exclusion. Molecular Ecology Notes 2006; 6: 265-267.
